# Tobacco Mild Green Mosaic Virus (TMGMV) Isolates from Different Plant Families Show No Evidence of Differential Adaptation to Their Host of Origin

**DOI:** 10.3390/v15122384

**Published:** 2023-12-05

**Authors:** Rafael de Andrés-Torán, Laura Guidoum, Adrian D. Zamfir, Miguel Ángel Mora, Santiago Moreno-Vázquez, Fernando García-Arenal

**Affiliations:** 1Centro de Biotecnología y Genómica de Plantas (CBGP UPM_INIA/CSIC), Universidad Politécnica de Madrid, Campus de Montegancedo, Pozuelo de Alarcón, 28223 Madrid, Spain; rafaelde.andres@upm.es (R.d.A.-T.); laurguid@ucm.es (L.G.); ad.zamfir@upm.es (A.D.Z.); miguelangel.mora@upm.es (M.Á.M.); 2Departamento de Biotecnología-Biología Vegetal, E.T.S.I. Agronómica, Alimentaria y de Biosistemas, Universidad Politécnica de Madrid, Avda. Puerta de Hierro 2-4, 28040 Madrid, Spain; santiago.moreno@upm.es

**Keywords:** tobamovirus, host range, host range evolution, phenotypic plasticity, ecological fitting, virus adaptation to host

## Abstract

The relevance of tobamoviruses to crop production is increasing due to new emergences, which cannot be understood without knowledge of the tobamovirus host range and host specificity. Recent analyses of tobamovirus occurrence in different plant communities have shown unsuspectedly large host ranges. This was the case of the tobacco mild green mosaic virus (TMGMV), which previously was most associated with solanaceous hosts. We addressed two hypotheses concerning TMGMV host range evolution: (i) ecological fitting, rather than genome evolution, determines TMGMV host range, and (ii) isolates are adapted to the host of origin. We obtained TMGMV isolates from non-solanaceous hosts and we tested the capacity of genetically closely related TMGMV isolates from three host families to infect and multiply in 10 hosts of six families. All isolates systemically infected all hosts, with clear disease symptoms apparent only in solanaceous hosts. TMGMV multiplication depended on the assayed host but not on the isolate’s host of origin, with all isolates accumulating to the highest levels in *Nicotiana tabacum*. Thus, results support that TMGMV isolates are adapted to hosts in the genus *Nicotiana*, consistent with a well-known old virus–host association. In addition, phenotypic plasticity allows *Nicotiana*-adapted TMGMV genotypes to infect a large range of hosts, as encountered according to plant community composition and transmission dynamics.

## 1. Introduction

Understanding the evolution of host range is a major goal in virology. Host range, or the number of host species exploited by a virus, may affect the genetic diversity of the virus and its ability to survive in the environment between epidemics in any one host. Host range determines the diversity of virus reservoirs and inoculum sources, and, by expanding to new hosts or shifting among hosts in response to environmental changes, disease emergence [1,2,3]. The genetics of host range evolution have been much studied, and both theoretical and experimental evidence predict that specialization in host traits would result in across-host fitness trade-offs, so that the increase in the virus fitness in one host would result in a fitness decrease in others [1,2,4,5,6,7]. However, there is also evidence indicating that viruses may expand their host ranges with little or no fitness costs, because phenotypic plasticity allows virus genotypes to colonize new hosts without adaptive evolution [8,9,10]. That is, ecological fitting [11] may play a role in virus–host range evolution.

Knowledge of plant virus host ranges in nature is incomplete, which is due in part to studies having focused on viruses that are pathogenic to crops or to wild plants growing in the crop environment [12,13,14]. High throughput sequencing (HTS) approaches are quickly changing this situation, as they allow the identification of plant–virus interactions regardless of whether they result in disease or not [15,16,17,18]. Reports of HTS studies of viromes in different plant communities, e.g., [19,20,21,22,23,24,25,26,27,28] reveal plant–virus associations that are often unexpected in light of previous knowledge on host ranges. One such study showed that four virus species of the genus Tobamovirus had unexpectedly large host ranges in a heterogeneous ecosystem [10].

Tobamoviruses (genus Tobamovirus, family Virgaviridae) are important pathogens of crops worldwide, and the relevance of species of this genus for agricultural production is increasing due to the recent emergence of new species [29,30,31]. Tobamoviruses have single-stranded, positive sense RNA genomes encapsidated in highly stable rod-shaped particles that allow long-term survival in the environment. Tobamoviruses are vertically transmitted though the seed, and horizontally by plant-to-plant contact [32,33] and by insect pollinators [34,35,36]. Tobamoviruses, particularly tobacco mosaic virus (TMV), have played a central role in the development of virology, including the field of virus evolution [37]. However, information on their ecology or the role of hosts other than crops in their epidemiology is scant. Tobamoviruses are considered to have narrow natural host ranges, and phylogenetic analyses strongly suggest that they have codiverged with the host plants in which they were first reported [38], which implies adaptation to hosts in the corresponding plant families. Recent reports from studies of wild plant communities in anthropic environments or near crop fields show that host ranges for different tobamoviruses are larger than previous knowledge indicated [22,23,39,40]. In a recent report [10], tobamovirus–plant interactions were analysed in four major habitats under different levels of human intervention in a complex agroecosystem of central Spain. Interactions were detected by HTS and confirmed by RT-PCR with virus-specific primers, which showed that the analysed virus species were found to be associated with host species in families unrelated to those of previously reported hosts. Large host ranges were not associated with genetic diversification of the studied viruses that, on the contrary, showed very limited genetic diversity, suggesting that ecological fitting had a main role in the determination of the observed host ranges [10]. The species with the largest host range was tobacco mild green mosaic virus (TMGMV), which was found to be associated with 73 host species of 23 families [10]. TMGMV had been reported previously mostly in species of the Solanaceae, but also of other families, such as Apiaceae, Asteraceae, Gesneriaceae and Scrophulariaceae [41,42] and, more recently, Cucurbitaceae [22,23]. TMGMV, first named as the U2 strain of tobacco mosaic virus (U2-TMV), was first isolated from *Nicotiana glauca* Grah. plants from the Canary Islands early in the XXth century [43], and was later reported infecting this wild plant in many world regions, including California, India, the Middle East, Australia and the Mediterranean basin [44,45]. Data on infection frequency, virus multiplication rates, and competitive ability with TMV, indicate that TMGMV is well adapted to *N. glauca* [45,46,47]. TMGMV has also been reported infecting tobacco crops in Germany since the 1940s, where at the end of the season, incidence reached 100%, causing big losses in burley tobacco production [42] and, later infecting pepper crops in California in the 1950s and in Italy in the 1980s [42]. Since then, TMGMV has emerged as an important pathogen of pepper crops in the Mediterranean basin [31,48], causing a highly severe syndrome of systemic necrosis [48].

Within this context, we experimentally test two nonexclusive hypotheses regarding the evolution of the TMGMV host range: (i) ecological fitting, rather than genome evolution, plays a role in determining the host range of this virus and (ii) isolates are adapted to the host of origin. To verify these hypotheses, we: (1) assay if TMGMV isolates from host species of a certain plant family can infect systemically host species from unrelated families, and (2) quantify the virus accumulation in inoculated and in systemically infected leaves.

## 2. Materials and Methods

### 2.1. Virus Isolates

Five isolates of TMGMV were used. Isolates 96/5 and 92/73, from *N. glauca* (wild tobacco, Solanaceae), and P92/10 from pepper (*Capsicum annuum* L., Solanaceae) were randomly chosen among those described in [49]. New isolates, PV063 from melon (*Cucumis melo* L. Cucurbitaceae) and PV081 from *Lithospermum arvense* L. (Boraginaceae), were obtained by inoculating in the local lesion host *Nicotiana tabacum* L. cv. Xanthi-nc RNA extracted from plants of *Cucumis melo* L. (melon) and *L. arvense*, respectively. RNA extracts were from individual plants and had been pooled for HTS, yielding libraries PV063 and PV081, which were positive for TMGMV by both HTS and RT-PCR detection [10]. A single necrotic local lesion that developed in tobacco Xanthi-nc was transferred to tobacco cv. Samsun for a single step of virus multiplication. Particles of all TMGMV isolates were purified from Samsun tobacco leaves and virion RNA was extracted by phenol-chloroform after virus disassembly in 10% SDS at 65 °C as in [50].

### 2.2. Assayed Plant Hosts and Inoculations

Virus infection was assayed in 10 host plant species from six families: *N. tabacum* cv. Samsun (tobacco, Solanaceae), *C. annuum* L. cv. Dulce Italiano (pepper, Solanaceae), *Cucumis melo* L. cv. Piel de Sapo and cv. Amarillo (melon, Cucurbitaceae), *Carduus assoi* Willk. (Asteraceae, BGV7002), *Anchusa undulata* L. (Boraginaceae, BGV6996), *Centaurea paniculata* L. (Asteraceae, BGV4347), *Marrubium supinum* L. (Lamiaceae, BGV4242), *Silybum eburneum* Coss & Durieu (Asteraceae, BGV8864), *Hordeum vulgare* L. cv. Golden Promise (barley, Poaceae) and *Brachypodium distachyon* L. (Beauv.) (Poaceae). Seeds of crops and of the model monocot *B. distachyon* were obtained from the FGA collection, while seeds from wild germplasm came from Banco de Germoplasma Vegetal “César Gómez Campo”, Universidad Politécnica de Madrid (ES003). Hosts were chosen with three criteria: (i) being well known, economically relevant hosts of TMGMV (*C. annuum* and *N. tabacum*) from which field isolates have been well characterised, (ii) belonging to the three families with more TMGMV hosts in [10], i.e., Asteraceae (three species), Poaceae (two species) and Lamiaceae (one species) and (iii) belonging to the family of the hosts of the two new isolates, Cucurbitaceae and Boraginaceae.

Seeds were germinated in Petri dishes and seedlings were transplanted to individual 1.5 L pots. Plants were grown in a P1-level containment greenhouse at 23–25 °C, 16 h light. Two weeks after transplant, plants were inoculated by rubbing 500 ng of virions into the three first expanded leaves, after dusting with carborundum. Leaf samples (3 discs, 5 mm diameter, taken from three different leaves) for virus detection and quantification were taken from inoculated leaves 7 days post-inoculation (dpi) and from upper, noninoculated leaves, 21 dpi. Because tobamoviruses are highly contagious, pots were placed in the greenhouse so that plants could not make contact even at the end of the experiments, and extreme caution was taken during watering to avoid spillover among pots/plants.

### 2.3. Nucleotide Sequence Determination and Analyses

The nucleotide sequence of the genome of the new TMGMV isolates PV063 and PV081 and of isolates 92/10 and 96/5, was determined by Sanger sequencing of RT-PCR amplicons of about 800 nucleotides (nt) that overlapped about 200 nt with adjacent ones. These amplicons were obtained by RT-PCR using primers designed on the TMGMV sequences in the databases, that are specified in Table A1. Sequencing was outsourced at StabVida, Lisbon, Portugal. The accession number for the genomic sequences of isolates 92/10, 96/5, PV063 and PV081 are SAMN38369919 to SAMN38369922, respectively.

Multiple sequence alignments of the studied sequences using ClustalW and nucleotide diversities (π) were computed in MEGA version 11 [51]. Maximum likelihood phylogenies relating TMGMV isolates were obtained by MEGA version 11 using Tamura-Nei values, and with 100 bootstrap replicates.

### 2.4. Quantification of Infectivity, Virus Multiplication and Virulence

Total RNA was extracted from 100 mg of leaf tissue using NZYol^®^ (NZYTech, Lisboa, Portugal). The status, infected or noninfected, of inoculated plants was assessed by RT-PCR using primers P0b-Q fwd (5′ GCACCGAATACTACT GAAATCG 3′) and P0-Q rev (5′ GCCTGCTTGATTGAA CATGCCAGT 3′) complementary and identical, respectively, to positions 5970–5991 and 6075–6098 of TMGMV-Jap genomic RNA (Accession No. AB078435). Amplicons were analysed by agarose electrophoresis in 1.2% TAE 1X (40 mM Tris pH 7.5, 20 mM sodium acetate, 2 mM EDTA). Virus accumulation was quantified by RT-qPCR using the same set of primers, and all samples were analysed in duplicates. Levels of viral RNA were deduced from comparison with standard curves generated using a set of serial dilutions of purified virion RNA [49].

### 2.5. Statistical Analyses

Data analyses always followed the same pipeline. First, outliers were identified using Grubbs’ test and removed. The distribution that best fit the data was evaluated by the Shapiro–Wilk test and, if data did not follow a normal distribution, QQ-plots, histograms, and probability curves were adjusted to the data to find the best-fit distribution, which was evaluated by Akaike’s Information Criterion (AIC). This process was performed as implemented in the R package rriskDistributions [52]. Distributions of virus accumulation were not normal, and were best fitted by Gamma and lognormal distributions, depending on the experiment. The homogeneity of variance was analysed using the Levene test. The effect of different factors on the analysed variables was analysed by General or Generalised Lineal Models (GLM), according to data distribution, and post hoc analysis for multiple comparisons (Fisher’s Least Significant Difference, Tukey’s test) was performed when applicable. All statistical analyses were performed using R (4.3.1) [53].

## 3. Results

### 3.1. Characterisation of Two New Isolates of TMGMV

The nucleotide sequence of the genome of isolates PV063 (6245 nt determined) and PV081 (6251 nt determined) was determined, with the exception of the 3′ untranslated region. The similarity between these new isolates, estimated as nucleotide diversity (π), was 0.012 ± 0.001. The similarity of the new isolates with two reference TMGMV isolates, (strain Japanese, TMGMV-Jap, AB078435.1, and isolate 92/73, TMGMV 92/73, MH730970.1) was also estimated. Similarity of PV063 and PV081 with TMGMV-Jap was π = 0.033 ± 0.002 and π = 0.038 ± 0.003, respectively; and similarity of PV063 and PV081 with TMGMV 93/73 was π = 0.010 ± 0.001 and π = 0.047 ± 0.001, respectively. Mutations in the genome of PV063 and PV081 relative to TMGMV-Jap were most frequent in the sequences encoding the methyltransferase domain of protein 183K (147 and 175 differences for PV063 and PV081, respectively), and the coat protein (CP) (29 and 28 differences for PV063 and PV081, respectively), while mutations relative to TMGMV 92/73 were more evenly distributed over the genome with the highest frequency in the CP gene (21 and 20 differences, respectively). The nucleotide sequence of the genome of isolates 96/5 (5997 nt determined) and P92/10 (5999 nt determined) was also determined, with the exception of the 5′ and 3′ untranslated regions. The similarity between these isolates, estimated as nucleotide diversity (π) was 0.010 ± 0.002. Similarity of 96/5 and P92/10 with TMGMV-Jap was π = 0.032 ± 0.003 and π = 0.034 ± 0.002, respectively; similarity of 96/5 and P92/10 with TMGMV 92/73 was π = 0.009 ± 0.001 and π = 0.011 ± 0.001, respectively. The similarity of 96/5 and P92/10 with PV063 was π = 0.002 ± 0.0006 and π = 0.009 ± 0.001, respectively; and with PV081 π = 0.014 ± 0.002 and π = 0.012 ± 0.002, respectively. Mutations in the genome of 96/5 and P92/10 relative to TMGMV-Jap were most frequent in the sequences encoding the 126K protein (70 and 74 differences for 96/5 and P92/10, respectively) and the coat protein (CP) (16 and 23 differences for 96/5 and P92/10, respectively), while mutations relative to TMGMV 92/73 were more evenly distributed over the genome with the highest frequency in the CP gene (8 and 15 differences, respectively).

The phylogenetic relationships of isolates PV063, PV081, 96/5 and P92/10 with other TMGMV isolates for which genomic sequences are reported in the databases showed that the four isolates were very similar to all other isolates and that they clustered with a *N. glauca* isolate from Spain (Figure 1). Thus, isolates PV063 and PV081 were most related to other isolates from Spain. Otherwise, TMGMV isolates did not cluster according to host or geographical region of origin.

### 3.2. Assay of the Host Range of Four TMGMV Isolates

The ability to infect different hosts of TMGMV isolates 96/5 from *N. glauca* (wild tobacco), P92/10 from *C. annuum* (pepper), PV081 from *L. arvense* and PV063 from *C. melo* (melon), was tested by inoculating each isolate into plants of 10 species of six botanical families (Table 1). The experiment involved 40 infection treatments (four isolates × 10 hosts) plus 10 mock-inoculated controls, with 10 replicated plants per treatment or controls. Treatments were randomised in the greenhouse. Because of poor germination, or early death of seedlings, some treatments of *C. paniculata* had nine replicas, while those of *A. undulata* and *S. eburneum* had three to six replicas, pending the treatment (Table 1). RT-PCR detection of TMGMV RNA in upper, noninoculated leaves was used to estimate infectivity. All species were infected by all isolates, except for *C. melo*, which was not infected by isolate 96/5, due to a failure during the process of inoculation. Thus, this treatment was not included in analyses of infectivity. A GLM analysis of the data (Table 1), assuming a binomial distribution and considering virus isolate and host species as fixed factors and their interaction, indicated that both virus isolate (F_3,277_ = 4.52; *p* = 0.004), host species (F_9,277_ = 4.81; *p* < 10^−3^), and their interaction (F_26,277_ = 2.23; *p* = 0.001), had a significant effect on infectivity. The effect of virus isolate was due to the lower infectivity of P081 relative to P063 (*p* = 0.025), while the infectivity of isolates 96/5, P92/10 and P063 did not differ (*p* > 0.073). The fraction of infected plants was higher in *N. tabacum* than in *C. assoi*, *C. melo*, *C. paniculata* or *M. supinum*, and in *B. distachyon* than in *C. assoi* and *C. melo* (*p* < 0.041), and did not differ for other binary comparisons (*p* > 0.137). Infectivity of P063 was consistently high in all hosts, while for the other isolates infectivity was lower in some hosts. Infectivity in the different hosts differed for each isolate, e.g., for 96/5 in *C. assoi* and *S. eburneum*, and for PV081 in *C. melo*, *C. paniculata* and *M. supinum* (Table 1).

Infected plants of *B. distachyon*, *C. melo*, *C. assoi*. *C. paniculata*, *H. vulgare* and *M. supinum*, did not show any symptoms. Infected plants of *A. undulata* and *S. eburneum* showed a light yellowing, those of *C. annuum* showed necrotic areas in leaves and necrotic streaks in stems and petioles, and those of *N. tabacum* showed a green mosaic. In no host did symptoms differ according to the infecting isolate.

### 3.3. Accumulation of TMGMV Isolates in Ten Host Plant Species

In the infected plants of the experiment described in the previous section, the accumulation of the four TMGMV isolates was quantified by RT-qPCR in the inoculated leaves and in upper, systemically infected leaves.

A GLM with virus isolate and host species as fixed factors, and their interaction showed that virus accumulation in the inoculated leaves depended on the virus isolate (F_3,189_ = 4.68; *p* = 0.003), on the host species (F_9,189_ = 52.68; *p* < 10^−3^) and on their interaction (F_26,189_ = 4.12; *p* < 10^−3^). Across hosts, isolates ranked according to their accumulation in the inoculated leaves as 96/5 = PV063 = PV081 > P92/10 (*p* < 10^−3^). Accumulation of the four virus isolates over host species ranked *N. tabacum* > *C. annuum* > *M. supinum* = *S. eburneum* > *A. undulata* = *C. paniculata* > *C. assoi* = *C. melo* > *B. distachyon* = *H. vulgare* (*p* < 0.048). For all isolates, accumulation was highest in *N. tabacum* and lowest in *B. distachyon* and *H. vulgare*, with accumulation levels varying for each isolate in the rest of the host species (Table 2).

Next, virus accumulation in systemically infected leaves was analysed. A GLM with virus isolate and host species as fixed factors, and their interaction showed that virus accumulation in systemically infected leaves depended on the host species (F_9,196_ = 119.61; *p* < 10^−3^) and the interaction virus isolate per host species (F_26,189_ = 2.43; *p* < 10^−3^) but not on the virus isolate (F_3,196_ = 0.76; *p* = 0.516). Considering the four virus isolates together, virus accumulations across hosts ranked *N. tabacum* > *C. annuum* = *C. paniculata* > *A. undulata* = *C. assoi* = *M. supinum* = *S. eburneum* > *C. melo = B. distachyon* > *H. vulgare* (*p* < 0.017). Accumulation was highest for all isolates in *N. tabacum* and lowest in *H. vulgare*, and differential interactions of isolates across hosts were more subtle than for inoculated leaves (compare Table 2 and Table 3). It is to be underscored that, across virus isolates and host species, virus accumulation was higher in inoculated than in systemically infected leaves. It is also worth noting that hosts did not rank the same for virus accumulation in inoculated and systemically infected leaves, although for both types of leaves accumulation was highest in *N. tabacum* and lowest in *H. vulgare*.

Because the inoculation of TMGMV 96/5 in *C. melo* failed, and because infection of TMGMV in this important crop may be potentially relevant and has been reported from different regions [23,24], the interaction between *C. melo* and TMGMV was further explored. For this, plants of *C. melo* from two cultivars commonly grown in Spain that have broadly different genetic backgrounds, Amarillo and Piel de Sapo, were inoculated with three TMGMV isolates: PV063 from *C. melo*, 96/5 from *N. glauca*, and 92/73 also from *N. glauca*. The inclusion of this second isolate from *N. glauca* was because, in a preliminary experiment, this isolate became systemic in four out of six Piel de Sapo inoculated plants, as assessed by RT-PCR (not shown). The experiment had six randomised treatments plus mock inoculated controls and five to eight replicated plants per treatment or controls. All inoculated plants became systemically infected except for isolate 92/73 that infected 7/8 Piel de Sapo plants. A GLM of virus accumulation in the inoculated leaves 7 dpi (Table 4) with virus isolate and host cultivar as fixed factors plus the interaction virus isolate per host cultivar showed that accumulation did depend on the host cultivar (F_1,24_ = 5.63; *p* = 0.024), but not on the virus isolate nor the virus isolate per host cultivar interaction (F_2,24_ = 1.01; *p* = 0.377; F_2,24_ = 0.33; *p* = 0.724). Accumulation was higher in Amarillo than in Piel de Sapo. A GLM of virus accumulation in systemically infected leaves 21 dpi (Table 4) with virus isolate and host cultivar as fixed factors plus the interaction virus isolate x host cultivar showed that virus accumulation did not depend on the host cultivar (F_1,23_ = 1.16; *p* = 0.290) but depended on the virus isolate (F_2,23_ = 5.72; *p* = 0.008) and on the interaction virus isolate per host cultivar (F_2,23_ = 6.56; *p* = 0.005). TMGMV isolate PV063, from *C. melo* cv. Piel de Sapo, accumulated to higher levels than the two isolates from *N. glauca*, 96/5 and 92/73 (*p* < 0.011), and the significance of the interaction virus isolate per host cultivar was due to the higher accumulation of PV063 in Amarillo, as compared with the other interactions (Table 4). The results of this experiment were consistent with those involving 10 host species, in which, despite detection of viral RNA in higher noninoculated leaves by RT-PCR, systemic accumulation of TMGMV in *C. melo* was extremely low. In addition, it unveiled that systemic accumulation was higher for isolate PV063, which shared its host of origin with the cultivar-assayed host species.

To confirm the presence of infectious virus in upper noninoculated leaves, extracts from upper leaves of plants where the virus was detected by RT-PCR were used to inoculate leaves of *N. tabacum* cv. Xanthi-nc. The assay involved 12 plants of *N. tabacum, C. assoi* and *C. paniculata*, and 25 plants of *C. melo*, from both assays involving this host, that is, the assay involved hosts where TMGMV accumulation in upper leaves was high, low or very low. The presence of infectious virus was shown by the appearance of necrotic local lesions (nll) 7 dpi, which occurred in inoculations from all *N. tabacum* plants, 83% of *C. assoi*, 50% of *C. paniculata* and 32% of *C. melo* plants.

## 4. Discussion

Knowledge of the host range of viruses is necessary to fully understand virus ecology, epidemiology and evolution [2,7,54]. However, knowledge of virus–host ranges is necessarily incomplete as host range is a plastic phenotype that is determined by the distribution, abundance, and interaction with potential hosts [2]. Gaps in the knowledge of the host range of plant viruses are large, because traditionally, studies have focused on viruses that are pathogenic to crops or, more recently, to wild plants, and viruses often are not pathogenic to wild plants and crops, or even behave as facultative mutualists; thus, they go unnoticed [14,55]. HTS allows the detection of virus–plant interactions with no bias towards those resulting in disease [15,16,17,18], which is quickly changing views on the diversity of virus–plant interactions. Some of us have recently reported, based on HTS detection validated by RT-PCR with specific primers, that tobamoviruses have broad host ranges in a heterogeneous ecosystem of central Spain [10]. TMGMV showed the broadest host range, with hosts in 23 families of mono and dicotyledon plants, a result inconsistent with knowledge on the host range of TMGMV, which has been reported primarily in hosts from the Solanaceae [38,41,42]. On the other hand, these findings were consistent with the very broad experimental host ranges of tobamoviruses [56,57,58]. Infection of host species of mono- and dicotyledon families by highly similar TMGMV genotypes suggested that infection of new hosts did not require genetic adaptation involving across-host fitness trade-offs. It was proposed that even if tobamoviruses had evolved associated primarily with hosts of one family, as phylogenetic evidence indicates [38], phenotypic plasticity with no or small across-host fitness trade-offs, and a variety of encounters with new potential hosts would explain the broad host ranges found in nature.

In this work, we address these hypotheses by quantifying the infectivity and accumulation of four TMGMV isolates from different hosts in hosts from the family of their host of origin or from other families. For this, we obtained TMGMV isolates from two new hosts among those reported in [10]: isolate PV063 from the cultivated melon (*C. melo*) and isolate PV081 from the wild *L. arvense*. *C. melo* was chosen as the hosts of origin because of the potential relevance of TMGMV in this important crop, as HTS analyses of virus infections in melon crops and nearby wild plant communities had already detected TMGMV infection of *C. melo* in different regions of Spain [23,24], and melon had been confirmed experimentally as a host of TMGMV [23]. *L. arvense* was chosen at random among hosts in families not previously reported as hosts of TMGMV. Consistent with the analyses of HTS-derived partial genome sequences in [10], the Sanger sequence of the genomes of PV063 and PV081 are highly similar to those of TMGMV from other hosts and, particularly, with those from isolates from the same geographical region, in support of a role of ecological fitting in TMGMV host range determination. However, it has been shown that the differential fitness of tobamovirus genotypes in different hosts may depend on a few mutations [49,59], which prompted us to quantify two fitness components, infectivity and within-host multiplication, of four TMGMV isolates in a panel of 10 hosts from six botanical families. As recently collected seeds of wild plants may be difficult to use in experimentation due to dormancy, we used seeds from plants species in the same genus as major TMGMV hosts identified in [10], and from the same geographical area, which were available at the germplasm bank César Gómez Campo of Universidad Politécnica de Madrid.

Detection of the four TMGMV isolates in upper noninoculated leaves indicates they infected systemically each of the 10 assayed hosts. Although we cannot rule out that presence of virus in upper leaves was due to cross-contamination among plants, it is unlikely that this possibility explains the high rates of systemic infection observed (Table 1), considering the measures taken to avoid plant-to-plant contact, spillover during watering, and the way tissue samples were taken from leaves. Also, it could be that in the hosts in which systemic multiplication was less efficient, virus multiplication in upper leaves weas due to cell-to-cell rather than to phloem movement. We cannot test this hypothesis, as an analysis of the mechanisms of host colonization was not an objective of this study, but the relevant fact for the epidemiology and host range evolution of the virus is that the virus was present in, and could be transmitted from, upper leaves. Interestingly, clear symptoms of disease were observed only in *C. annuum* and *N. tabacum*, two hosts in which TMGMV is known for a long time as an important crop pathogen [42,48]. In most other hosts, infection was asymptomatic, suggesting TMGMV did not have deleterious effects on these hosts, as in virus–plant interactions it has been shown that the deleterious effects of infection correlate with symptom expression [60]. Nonpathogenicity is probably also the situation in natural field conditions, as the pathogenicity of plant viruses has been shown to change towards a conditional mutualism in conditions stressful for the plant [61,62] that will occur more frequently under the harsh conditions of the Mediterranean climate of central Spain than in the greenhouse. Infectivity, estimated as the fraction of inoculated plants that became systemically infected, depended on the interaction virus isolate per host species (Table 1). This is an interesting result, as virus inoculation was carried at a very high inoculum dose, which was expected to saturate infection sites. However, the differential infectivity of each isolate across hosts was no evidence of host adaptation for this trait, as infectivity was not higher in the host or family of the host of origin of the isolate. Rather, all isolates had a 90–100% infectivity in *N. tabacum*, which did not occur in any other host.

Within-host virus multiplication was estimated by quantifying virus accumulation in both the inoculated and the systemically infected leaves at two times after infection. Virus accumulation also depended on the interaction virus isolate per host species, but while host species had an effect on the accumulation in both leaf types, an effect of virus isolate was only detected in the inoculated leaves. This, again, is an interesting result, as accumulation in systemically infected leaves may better reflect the nature of the interaction, the accumulation in the inoculated leaves being more dependent on the artificial conditions of mechanical inoculation in the experiment [63]. In both the inoculated and the systemically infected leaves, accumulation of the four TMGMV isolates was highest in *N. tabacum*, regardless of the host of origin of the isolate. This result indicates that TMGMV isolates, regardless of host of origin, are best adapted to *N. tabacum* and, probably other *Nicotiana* species, than to other hosts, according to the “home vs. away” criterium for local adaptation [64]. *N. tabacum* and *N. glauca* are the two species of the genus *Nicotiana* where TMGMV was first reported, and in whose populations it is known to occur at a high incidence [42,44,45,65,66], indicating high host competence and suggesting a long virus–host association. Thus, adaptation of TMGMV to *N. tabacum* is fully compatible with the hypothesis of codivergence of tobamoviruses with their hosts derived from phylogenetic analyses [38]. In all other host species, virus accumulation was much lower and even extremely low, which is consistent with the low number of reads matching the TMGMV reference in most HTS libraries of *C. melo* and of hosts in the Asteraceae, Poaceae, Lamiaceae and Boraginaceae [10,24]. These low titres are also compatible with infection of hosts randomly encountered in different plant communities due to ecological fitting [11], as hypothesised [10], and as also proposed for aphid-transmitted viruses [8,9]. Of note is the extremely low accumulation of TMGMV in systemically infected leaves of *C. melo*, consistently shown in two cultivars challenged with five TMGMV isolates in two different experiments. The higher accumulation in cv. Amarillo of isolate PV063, from *C. melo* than of isolates 96/5 and 92/73, from *N. glauca*, leads to speculate on an incipient adaptation of TMGMV to *C. melo*, a hypothesis that would require further analyses. The efficiency of contact transmission of tobamoviruses depends on virus titre in the source leaf, among other factors, including leaf age [67]. Thus, the low levels of TMGMV accumulation in most assayed hosts suggest inefficient transmission from and among most hosts, and do not help to explain the reported widespread occurrence of TMGMV in both anthropic and wild plant communities of a heterogeneous ecosystem [10].

In summary, our present and past results allow us to draw a plausible scenario for the evolution of the host range of TMGMV: TMGMV seems to have had a long association with hosts in the genus *Nicotiana*, to which isolates from different hosts are best adapted. In addition, ecological fitting allows *Nicotiana*-adapted TMGMV genotypes to successfully infect systemically a large diversity of potential hosts, as encountered according to plant community composition and transmission dynamics. Infection of a large set of hosts with no need for genetic adaptation, followed by spatial compartmentation of infections [24] might, in turn, result in future adaptation to those hosts more frequently infected.

## Figures and Tables

**Figure 1 viruses-15-02384-f001:**
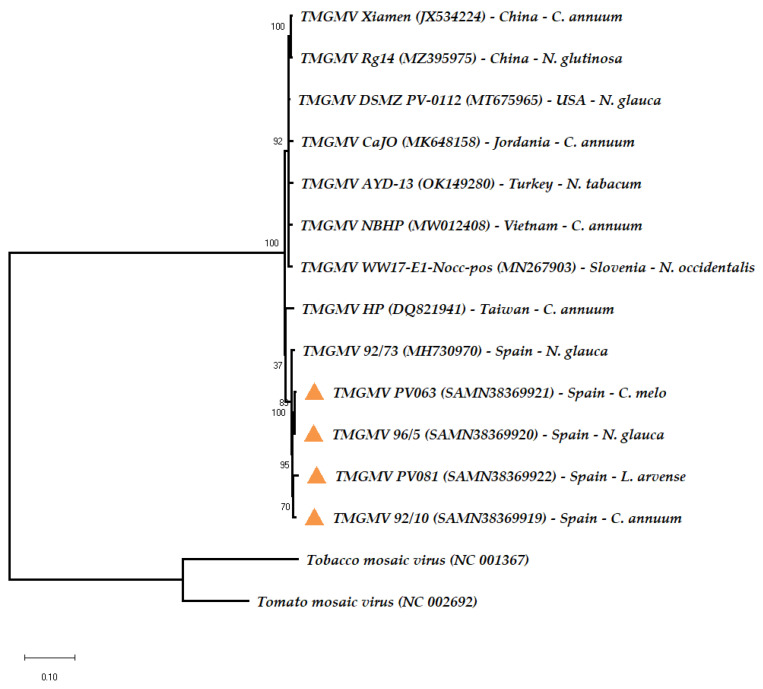
Maximum Likelihood phylogeny of 13 whole-genome sequences from TMGMV isolates from different hosts and regions, as indicated. Significance of nodes in a Bootstrap analysis with 100 pseudoreplicas is shown, and the tree with the highest log likelihood (−17,437.12) is shown. Sequences of tobacco mosaic virus (TMV) and tomato mosaic virus (ToMV) were used as outgroups. Orange triangles indicate the isolates whose genome sequence was determined in this work. 92/10 stands for isolate P92/10.

**Table 1 viruses-15-02384-t001:** Infectivity of four TMGMV isolates in 10 host plant species ^a^. The host of origin of each TMGMV isolate is indicated below the isolate’s name.

Host Species	Family	96/5 *N. glauca*	P92/10 *C. annuum*	PV081 *L. arvense*	PV063 *C. melo*	Total
*A. undulata*	Boraginaceae	5/5 (100)	4/4 (100)	4/5 (80)	3/5 (60)	16/19 (84)
*B. distachyon*	Poaceae	8/10 (80)	9/10 (90)	10/10 (100)	10/10 (100)	37/40 (92)
*C. annuum*	Solanaceae	9/10 (90)	10/10 (100)	6/10 (60)	10/10 (100)	35/40 (87)
*C. assoi*	Asteraceae	5/10 (50)	6/10 (60)	7/10 (70)	8/10 (90)	26/40 (65)
*C. melo* cv. Piel de Sapo	Cucurbitaceae	0/10 (0)	8/10 (80)	4/9 (44)	9/9 (100)	21/38 (55)
*C. paniculata*	Asteraceae	7/10 (70)	5/10 (50)	5/9 (55)	9/9 (100)	26/38 (68)
*H. vulgare*	Poaceae	7/10 (70)	10/10 (100)	10/10 (100)	8/10 (80)	35/40 (87)
*M. supinum*	Lamiaceae	7/10 (70)	6/9 (67)	5/9 (55)	8/10 (80)	26/38 (68)
*N. tabacum*	Solanaeae	10/10 (100)	9/10 (90)	10/10 (100)	10/10 (100)	39/40 (98)
*S. eburneum*	Asteraceae	3/5 (60)	4/5 (80)	4/6 (67)	5/6 (83)	16/22 (73)
Total	-	61/90 (68)	71/88 (81)	65/88 (74)	80/89 (90)	277/355 (78)

^a^ Data are the number of infected plants over inoculated plants, with percentage of infection between parentheses.

**Table 2 viruses-15-02384-t002:** Accumulation of four TMGMV isolates (ng viral RNA/μg total RNA) in inoculated leaves of 10 host plant species ^a^. The host of origin of each TMGMV isolate is indicated below the isolate’s name.

Host Species	Family	96/5 *N. glauca*	P92/10 *C. annuum*	PV081 *L. arvense*	PV063 *C. melo*	Total
*A. undulata*	Boraginaceae	0.040 ± 0.011	0.317 ± 0.165	4.264 ± 4.182	4.290 ± 4.253	2.230 ± 1.026
*B. distachyon*	Poaceae	0.047 ± 0.013	0.058 ± 0.016	0.035 ± 0.009	0.032 ± 0.009	0.038 ± 0.008
*C. annuum*	Solanaceae	27.450 ± 8.881	0.142 ± 0.049	36.610 ± 8.334	28.350 ± 6.147	23.100 ± 6.882
*C. assoi*	Asteraceae	0.072 ± 0.031	0.116 ± 0.046	0.002 ± 0.001	1.524 ± 0.663	0.428 ± 0.317
*C. melo* cv. Piel de Sapo	Cucurbitaceae	-	0.383 ± 0.077	0.258 ± 0.098	0.113 ± 0.026	0.251 ± 0.055
*C. paniculata*	Asteraceae	0.073 ± 0.039	0.547 ± 0.480	0.104 ± 0.094	7.921 ± 4.302	2.160 ± 1.665
*H. vulgare*	Poaceae	0.041 ± 0.027	0.041 ± 0.011	0.020 ± 0.011	0.032 ± 0.022	0.028 ± 0.007
*M. supinum*	Lamiaceae	11.630 ± 4.769	0.023 ± 0.007	7.791 ± 3.111	5.062 ± 1.996	6.130 ± 2.110
*N. tabacum*	Solanaeae	63.782 ± 5.520	32.429 ± 7.202	53.686 ± 3.463	40.773 ± 2.178	47.700 ± 5.998
*S. eburneum*	Asteraceae	12.725 ± 6.377	0.136 ± 0.037	10.922 ± 10.841	15.631 ± 9.575	9.850 ± 2.929
Total	-	12.873 ± 7.098	3.419 ± 3.224	11.369 ± 5.896	10.373 ± 4.402	9.191 ± 4.838

^a^ Data are mean and standard errors of a number of replicates equal to that of infected plants in Table 1, after removal of outliers.

**Table 3 viruses-15-02384-t003:** Accumulation of four TMGMV isolates (ng viral RNA/μg total RNA) in systemically infected leaves of 10 host plant species ^a^. The host of origin of each TMGMV isolate is indicated below the isolate’s name.

Host Species	Family	96/5 *N. glauca*	P92/10 *C. annuum*	PV081 *L. arvense*	PV063 *C. melo*	Total
*A. undulata*	Boraginaceae	0.1564 ± 0.1538	0.0015 ± 0.0006	0.0011 ± 0.0004	2.3508 ± 2.3498	0.627 ± 0.498
*B. distachyon*	Poaceae	0.0004 ± 0.0002	0.0003 ± 0.0002	0.0003 ± 4 × 10^−5^	0.0014 ± 0.0003	0.0006 ± 0.0002
*C. annuum*	Solanaceae	0.2604 ± 0.1238	0.2118 ± 0.0714	0.2795 ± 0.2578	0.0075 ± 0.0018	0.1900 ± 0.0540
*C. assoi*	Asteraceae	0.0060 ± 0.0026	0.0032 ± 0.0011	0.0014 ± 0.0005	0.0112 ± 0.0056	0.0054 ± 0.0019
*C. melo* cv. Piel de Sapo	Cucurbitaceae	-	0.00024 ± 6.3 × 10^−5^	0.0004 ± 0.0001	6.6 × 10^−5^ ± 8 × 10^−6^	0.0002 ± 0.00006
*C. paniculata*	Asteraceae	0.0865 ± 0.0223	0.0819 ± 0.0182	0.0608 ± 0.0093	0.0284 ± 0.0061	0.0644 ± 0.0115
*H. vulgare*	Poaceae	9.1 × 10^−5^ ± 2.1 × 10^−5^	0.0003 ± 0.0001	0.0001 ± 7.5 × 10^−5^	0.0002 ± 5.9 × 10^−5^	0.0002 ± 0.00005
*M. supinum*	Lamiaceae	0.0018 ± 0.0003	0.0006 ± 0.0001	0.0012 ± 0.0004	0.0007 ± 9 × 10^−5^	0.0011 ± 0.0002
*N. tabacum*	Solanaeae	16.6255 ± 7.3280	10.3038 ± 4.5800	22.5143 ± 5.6892	38.9237 ± 11.0730	22.1000 ± 5.3170
*S. eburneum*	Asteraceae	0.0035 ± 0.0017	0.0036 ± 0.0013	0.0113 ± 0.0055	0.0151 ± 0.0079	0.0084 ± 0.0025
Total	-	1.905 ± 1.840	1.061 ± 1.027	2.287 ± 2.248	4.134 ± 3.873	2.2989 ± 2.2001

^a^ Data are mean and standard errors of a number of replicates equal to that of infected plants in Table 1, after removal of outliers.

**Table 4 viruses-15-02384-t004:** Accumulation of three TMGMV isolates (ng viral RNA/μg total RNA) in inoculated and systemically infected leaves of two C. melo cultivars ^a^. The host of origin of each TMGMV isolate is indicated below the isolate’s name.

Assayed Leaf	Host	Cultivar	96/5 *N. glauca*	92/73 *N. glauca*	PV063 *C. melo*
Inoculated	*C. melo*	Amarillo	0.068 ± 0.016	0.237 ± 0.133	0.114 ± 0.022
Inoculated	*C. melo*	Piel de Sapo	0.038 ± 0.016	0.133 ± 0.057	0.055 ± 0.02
Systemically infected	*C. melo*	Amarillo	0.0002 ± 10^−5^	0.0003 ± 10^−5^	0.0009 ± 0.0004
Systemically infected	*C. melo*	Piel de Sapo	0.0003 ± 10^−5^	0.0003 ± 10^−5^	0.0003 ± 10^−5^

^a^ Data are mean and standard errors of five replicates for Amarillo and 5–7 replicates for Piel de Sapo, after removal of outliers.

## Data Availability

Data is contained within the article.

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
