# Peer review of "Tobacco Mild Green Mosaic Virus (TMGMV) Isolates from Different Plant Families Show No Evidence of Differential Adaptation to Their Host of Origin"

_viruses, 2023, doi:10.3390/v15122384_

Round 1

Reviewer 1 Report

Comments and Suggestions for Authors

This manuscript dealt with coevolution between host and viris which is one of main topics in virology. 

There is no correction and well designed and written for publication.

Author Response

  1. Thank you for these positive comments.

Reviewer 2 Report

Comments and Suggestions for Authors

This is a systematic demonstration that the tobamovirus, tobacco mild green mosaic virus

 (TMGMV), while being adapted for solanaceous hosts, can infect and move systemically in a wide range of non-solanaceous hosts, albeit to quite low levels.  The ability of TMGMV to replicate and move systemically in a broad range of wild hosts, without specific adaptation to those hosts, could be an important feature of this virus to maintain itself and spread in the environment in the absence of solanaceous hosts and represent a step on the way to adaptation.

With the advent of HTS, this piece of work nicely summarises and explains the prevalence of TMGMV in wild plants and extends our understanding of the ecology of tobamoviruses.  As such it will be of interest to all plant virologists.

Points to address:

Ln239. “ Across hosts, isolates ranked according to their accumulation in  the inoculated leaves as 96/5 = PV063 > P92/10 = PV081 (P < 0.046)”

It looks odd that 96/5 = PV063 even though PV081 has a higher average virus accumulation across hosts than PV063.

Ln 240. “Accumulation of the  four virus isolates over host species” ranking doesn’t seem to match how the data looks by eye.  Eg  S. eburneum, looks like one of the hosts allowing higher viral titres in inoculated leaves, yet is ranked = to B. diatachyon.

Ln 243.  “For all isolates, accumulation was highest in N. tabacum and lowest in C. melo and  H. vulgare, “

Looking at Table 2, B. distachyon looks like it has similarly low levels of virus accumulation to

H. vulgare , rather than C.melo having the lowest level of virus accumulation. 

Minor points/typos

Could the authors mention what cultivar of melon was used in Table 2 and 3.

Ln 113 Virus infection was assayed in ten hosts plant species from six families: N. tabacum:  Replace hosts with host.

Ln 141 Accession number for the genomic sequences of PV063 and PV081 are XXXX and YYYY, respectively, and for the CP gene of 96/5 and P92/19 VVVV and ZZZZ. : Insert the correct accession numbers.

Ln 182.  183K (53/244) and the coat 182 protein (CP) ((29/244), while mutations relative to TPMGV 92/73 were more evenly dis tributed over the genome with the highest frequency in the CP gene 92/73 (21/110) . : Explain what the numbers in brackets refer to eg 29/244.

Ln 230:  In no host symptoms differed according to the infecting isolate. Should this be non-host?

Ln 354.  In most other hosts infection was asymptomatic, suggesting TMGMV had not  deleterious effects on these hosts, :  suggesting that TMGMV did not have  deleterious effects on these hosts,

Comments on the Quality of English Language

Quality of English is high

Author Response

This is a systematic demonstration that the tobamovirus, tobacco mild green mosaic virus (TMGMV), while being adapted for solanaceous hosts, can infect and move systemically in a wide range of non-solanaceous hosts, albeit to quite low levels.  The ability of TMGMV to replicate and move systemically in a broad range of wild hosts, without specific adaptation to those hosts, could be an important feature of this virus to maintain itself and spread in the environment in the absence of solanaceous hosts and represent a step on the way to adaptation.

With the advent of HTS, this piece of work nicely summarises and explains the prevalence of TMGMV in wild plants and extends our understanding of the ecology of tobamoviruses.  As such it will be of interest to all plant virologists.

We are happy that the reviewer finds this is an interesting study.

Points to address:

Ln239. “ Across hosts, isolates ranked according to their accumulation in  the inoculated leaves as 96/5 = PV063 > P92/10 = PV081 (P < 0.046)”

It looks odd that 96/5 = PV063 even though PV081 has a higher average virus accumulation across hosts than PV063.

We thank the reviewer for drawing our attention to this big mistake. Both the data in Table 2 and the statistical analyses reported were wrong, due to inefficient copying from the original files. These errors are now mended, and data in Table 2 are consistent with the reported analyses.

Ln 240. “Accumulation of the four virus isolates over host species” ranking doesn’t seem to match how the data looks by eye.  Eg  S. eburneum, looks like one of the hosts allowing higher viral titres in inoculated leaves, yet is ranked = to B. diatachyon.

 Ln 243.  “For all isolates, accumulation was highest in N. tabacum and lowest in C. melo and  H. vulgare, “ Looking at Table 2, B. distachyon looks like it has similarly low levels of virus accumulation to H. vulgare , rather than C.melo having the lowest level of virus accumulation.

Again, we thank the reviewer for drawing our attention to these inconsistencies in the presentation of data on virus accumulation in the inoculated leaves. What we wrote in the original version of the paper was wrong. We re-checked the original files and statistical analyses and now the correct ranking of hosts according to virus accumulation is listed.

Minor points/typos

Could the authors mention what cultivar of melon was used in Table 2 and Information on C. melo cultivar added to Tables 1-3.

Ln 113 Virus infection was assayed in ten hosts plant species from six families: N. tabacum:  Replace hosts with host.

Done.

Ln 141 Accession number for the genomic sequences of PV063 and PV081 are XXXX and YYYY, respectively, and for the CP gene of 96/5 and P92/19 VVVV and ZZZZ. : Insert the correct accession numbers.

Accession numbers now provided. Note than in the revised version we present the genomic sequences of isolates P92/10 and 96/5, not just the coat protein gene sequences. As a result, the genomic sequences of these isolates have been included in the phylogeny presented in Figure 1, and Fig. S1, now redundant, has been deleted.

Ln 182.  183K (53/244) and the coat 182 protein (CP) ((29/244), while mutations relative to TPMGV 92/73 were more evenly dis tributed over the genome with the highest frequency in the CP gene 92/73 (21/110) . : Explain what the numbers in brackets refer to eg 29/244.

These are the number of mutations of each isolate with the reference genome, now explained. Also, values for isolates 96/5 an P92/10 have been added, and values have been corrected for some comparisons after rechecking.

Ln 230:  In no host symptoms differed according to the infecting isolate. Should this be non-host?

No, they are asymptomatic hosts as the virus replicated in inoculated and upper non-inoculated leaves.

Ln 354.  In most other hosts infection was asymptomatic, suggesting TMGMV had not  deleterious effects on these hosts, :  suggesting that TMGMV did not have  deleterious effects on these hosts,

Mended.

Reviewer 3 Report

Comments and Suggestions for Authors

The authors of this manuscript tested two non-exclusive hypotheses regarding the host range evolution of TMGMV: (1) ecological fitness, rather than genome evolution, determines the host range of TMGMV; and (2) TMGMV isolates are adapted to the host of origin.  To examine these hypotheses, the authors (i) assayed whether virus isolates in host species of a particular family could infect systemically host species of unrelated families; and (ii) quantified virus accumulation in both the inoculated and upper leaves. All isolates infected variable numbers of all hosts tested, although clear symptoms were found only in the solanaceous species.  The extent of virus accumulation depended on the host species tested, but not the hosts species from the tested isolate originated; all isolates accumulated to the highest level in Nicotiana tabacum.  Hence, the TMGMV isolates are adapted to members of the genus Nicotiana, although the isolates retain the ability to infect a large range of hosts, to variable and often limited extents. 

I think this work was well done and is appropriate for publication in Viruses. However, I think the authors need to first address an important issue. This is the very highly infectious nature of tobamoviruses makes it difficult to determine whether very limited systemic accumulation is due to poor systemic movement or contact infection between leaves during such activity as watering the plants, if done with a hose, rather than drip irrigation, which can lead to leaves brushing each other or leaf hairs being blown off one leaf and making contact with another, sufficient to result in local infection at the site if contact. is occurred.  Another factor is that rather than true system movement, with a 3-week period between inoculation and sampling of upper leaves, that is sufficient time for cell-to-cell movement to occur in the parenchyma layers of petioles to stem to petioles of upper leaves and into upper parenchyma cells of the leaf, with vascular movement occurring.  Finally, although I do not think this is necessary to verify experimentally at this stage, the authors might want to address whether the low levels of virus accumulation they saw in many instances was likely to represent extremely low levels of virus accumulation in many cells, or normal accumulation in a small number of cells, with highly restricted cell-to-cell movement.

As I recall, in ref. 56, Holmes with TMV and TEV that he could determine several types of hosts: (i) those that infected locally and systemically (+/- symptoms); (ii) those that could infect locally only (+/- local symptoms) with no virus recoverable from upper leaves by back inoculation to a test host: (iii) those with no symptoms and no virus recoverable form any leaves.  Are there are similarities between the TMGMV situation here vs. the TMV situation as described by Holmes for common hosts used in this study?

Comments on the Quality of English Language

There are some word choice issues that need to be fixed, but that probably can be done by the staff at Viruses. 

Author Response

The authors of this manuscript tested two non-exclusive hypotheses regarding the host range evolution of TMGMV: (1) ecological fitness, rather than genome evolution, determines the host range of TMGMV; and (2) TMGMV isolates are adapted to the host of origin.  To examine these hypotheses, the authors (i) assayed whether virus isolates in host species of a particular family could infect systemically host species of unrelated families; and (ii) quantified virus accumulation in both the inoculated and upper leaves. All isolates infected variable numbers of all hosts tested, although clear symptoms were found only in the solanaceous species.  The extent of virus accumulation depended on the host species tested, but not the hosts species from the tested isolate originated; all isolates accumulated to the highest level in Nicotiana tabacum.  Hence, the TMGMV isolates are adapted to members of the genus Nicotiana, although the isolates retain the ability to infect a large range of hosts, to variable and often limited extents. 

I think this work was well done and is appropriate for publication in Viruses. However, I think the authors need to first address an important issue. This is the very highly infectious nature of tobamoviruses makes it difficult to determine whether very limited systemic accumulation is due to poor systemic movement or contact infection between leaves during such activity as watering the plants, if done with a hose, rather than drip irrigation, which can lead to leaves brushing each other or leaf hairs being blown off one leaf and making contact with another, sufficient to result in local infection at the site if contact. is occurred.  The reviewer is totally correct: Having worked with tobamoviruses for more than 40 years we are well aware of the risks of cross-contamination due to direct or indirect contact among plants. Thus, pots were placed in the greenhouse at a distance that avoided plant-to-plant contact, and watering with a hose was done with extreme care to avoid spillover, watering being done on the substrate with no water falling on the plants. This is now explained in the M&M section (L134-137). Still, we cannot rule out that contamination occurred, resulting in contact infection of upper leaves, although from past experience and from data on the mock-inoculated controls we think that the probability was low. Also, the probability of this contact transmission being the cause of virus detection in upper leaves would be even smaller, as small disks of tissue were taken from three different leaves, which at least one should have been contaminated. Because detection in upper leaves occurred in a large fraction of inoculated plants, most probably it is not due mostly to contact transmission from other plants. We have added some sentences in the Discussion to this effect (L377-381).

Another factor is that rather than true system movement, with a 3-week period between inoculation and sampling of upper leaves, that is sufficient time for cell-to-cell movement to occur in the parenchyma layers of petioles to stem to petioles of upper leaves and into upper parenchyma cells of the leaf, with vascular movement occurring. 

Because the time needed for the virus to be systemic depends on the particular virus host interaction (e.g., in our greenhouse conditions 4 dpi for TMGMV in tobacco, 7-8 dpi for TMGMV in pepper plants) and we did not a priori knew if TMGMV would systemically infect eight of ten assayed hosts, and how efficiently, we chose a long time after inoculation to check for systemic infection. Thus, we cannot rule that the presence of the virus in upper leaves was due to cell-to-cell movement rather than to phloem transport, because our study was not set to analyse mechanisms of host colonisation. Still, from the perspective of host range evolution, we think the relevant trait is the ability to invade the host further than the inoculated leaves, and to be transmitted from the upper, non-inoculated leaves. To check this later possibility we did a local lesion assay that is now reported (L316-323) and discussed, and we got infections from all assayed hosts (four) with efficiencies related to the estimated amount of virus in upper leaves. We have added these points to the Discussion (L381-386).

Finally, although I do not think this is necessary to verify experimentally at this stage, the authors might want to address whether the low levels of virus accumulation they saw in many instances was likely to represent extremely low levels of virus accumulation in many cells, or normal accumulation in a small number of cells, with highly restricted cell-to-cell movement.

We agree with the reviewer that this interesting point is out of the scope of this work. However, we think that a low number of infected cells should be a factor in the low virus accumulation in upper leaves, as over hosts accumulation was higher in inoculated than in upper leaves. As we do not have data to test this issue, though, we think it is better not to address it in the text.

As I recall, in ref. 56, Holmes with TMV and TEV that he could determine several types of hosts: (i) those that infected locally and systemically (+/- symptoms); (ii) those that could infect locally only (+/- local symptoms) with no virus recoverable from upper leaves by back inoculation to a test host: (iii) those with no symptoms and no virus recoverable form any leaves.  Are there are similarities between the TMGMV situation here vs. the TMV situation as described by Holmes for common hosts used in this study?

Yes, there is a similarity, and we think that all hosts will belong to the first class, that is, hosts that are infected locally and systemically with or without symptoms, with the caution that virus in upper leaves could be due to contact transmission and not to colonisation, which, as discussed above, we think has a low probability.